# Structural basis of pathogen recognition by an integrated HMA domain in a plant NLR immune receptor

A Maqbool[1†], H Saitoh[2†], M Franceschetti[1], CEM Stevenson[1], A Uemura[2], H Kanzaki[2], S Kamoun[3], R Terauchi[2], MJ Banfield[1*]

[1]Department of Biological Chemistry, John Innes Centre, Norwich, United Kingdom; [2]Iwate Biotechnology Research Center, Kitakami, Japan; [3]The Sainsbury Laboratory, Norwich, United Kingdom

**Abstract** Plants have evolved intracellular immune receptors to detect pathogen proteins known as effectors. How these immune receptors detect effectors remains poorly understood. Here we describe the structural basis for direct recognition of AVR-Pik, an effector from the rice blast pathogen, by the rice intracellular NLR immune receptor Pik. AVR-PikD binds a dimer of the Pikp-1 HMA integrated domain with nanomolar affinity. The crystal structure of the Pikp-HMA/AVR-PikD complex enabled design of mutations to alter protein interaction in yeast and in vitro, and perturb effector-mediated response both in a rice cultivar containing Pikp and upon expression of AVR-PikD and Pikp in the model plant *Nicotiana benthamiana*. These data reveal the molecular details of a recognition event, mediated by a novel integrated domain in an NLR, which initiates a plant immune response and resistance to rice blast disease. Such studies underpin novel opportunities for engineering disease resistance to plant pathogens in staple food crops.

*For correspondence: Mark. banfield@jic.ac.uk

†These authors contributed equally to this work

Competing interests: The authors declare that no competing interests exist.

## Introduction

Plant diseases are a continuous threat to crop production and a major constraint on achieving food security. Rice blast disease, caused by the fungal pathogen *Magnaporthe oryzae*, is the biggest pre-harvest biotic threat to global rice production (*Pennisi, 2010*; *Dean et al., 2012*; *Liu et al., 2014*). This disease can cause the loss of enough rice to feed 212–742 million people annually (*Fisher et al., 2012*), and result in up to 100% yield loss in infected areas (*Dean et al., 2012*; *Liu et al., 2014*). The sustainability of rice production is critical, it is a staple food crop for greater than half the world's population.

Approaches to controlling blast disease have mainly been via the deployment of rice resistance (*R*) genes, which encode intracellular immune receptors known as NLRs (Nucleotide-binding, Leucine-rich-repeat (LRR) Receptors). NLRs are a conserved component of plants' innate immune systems and survey the host environment for perturbations caused by invading pathogens (*Dangl and Jones, 2001*; *Chisholm et al., 2006*; *Jones and Dangl, 2006*; *van der Hoorn and Kamoun, 2008*; *Dodds and Rathjen, 2010*). Most NLRs respond to the presence or activities of translocated pathogen effectors, proteins delivered by adapted pathogens to affect the physiology of the host to benefit the parasite (*Dodds and Rathjen, 2010*; *Win et al., 2012*; *Wirthmueller et al., 2013*). The recognition event often results in a robust immune response and localised cell death, which limits disease caused by biotrophic pathogens on their hosts. Most NLRs comprise a multi-domain architecture with central nucleotide-binding (NB-ARC) and C-terminal LRR regions. They also usually contain N-terminal coiled-coil (CC) or TOLL/interleukin-1 receptor (TIR) domains (*Takken and Goverse, 2012*). In at least some cases, NLRs function in pairs to deliver disease resistance, and these pairs can be tightly linked

**eLife digest** Plant diseases reduce harvests of the world's most important food crops including wheat, rice, potato, and corn. These diseases are important for both global food security and local subsistence farming. To fight these diseases, crops (like all plants) have an immune system that can detect the telltale molecules produced by disease-causing microbes (also known as pathogens) and mount a defence response to protect the plant.

Nucleotide-binding, leucine-rich repeat receptors (or NLRs for short) are plant proteins that survey the inside of plant cells looking for these telltale molecules. These receptors have played a central role in efforts to breed disease resistance into crop plants for decades, but little is known about how they work.

Maqbool, Saitoh et al. have now used a range of biochemical, structural biology and activity-based assays to study how one NLR from rice directly interacts with a molecule from the rice blast fungus. This fungus causes the most important disease of rice (called rice blast), and the fungal molecule in question is also known as an 'effector' protein. A technique called X-ray crystallography was used to reveal the three-dimensional structure of the effector bound to part of the NLR called the 'integrated HMA domain'. Biochemical techniques were then used to measure how strongly the effector (and other related effectors) interacted with this domain of the NLR.

These results, combined with a close examination of the three-dimensional structure, allowed a set of changes to be made to the effector that stopped it interacting with the NLR protein domain in the laboratory. Maqbool, Saitoh et al. then performed experiments in rice plants and showed that changes to the effector that stopped it interacting with the NLR domain also stopped the effector from triggering a defence response in plants. Similar results were also obtained in experiments that used the model plant *Nicotiana benthamiana*.

In the middle of the 20th century, an American plant pathologist called Harold Henry Flor proposed that the outcomes of interactions between plants and disease-causing microbes were based on interactions between specific biological molecules. The findings of Maqbool, Saitoh et al. provide a new structural basis for this model. A detailed picture of these molecular interactions will allow researchers to engineer tailored NLRs that detect a wider range of pathogen molecules. In the future such an approach could contribute to efforts to protect the world's most important crops from plant diseases.

genetically (*Eitas and Dangl, 2010*; *Cesari et al., 2014*; *Le Roux et al., 2015*; *Sarris et al., 2015*). The protein:protein interactions that underlie NLR pair function are starting to be elucidated (*Cesari et al., 2014*; *Williams et al., 2014*), but many unknowns remain.

Interestingly, most NLR pairs studied to date use one of the NLRs to detect the presence of specific effectors by direct binding (*Kanzaki et al., 2012*; *Cesari et al., 2013*; *Williams et al., 2014*; *Zhai et al., 2014*). One mechanism by which this can be achieved is via unconventional integrated domains in the NLRs (known as integrated decoy or sensor domains) that show evolutionary relationships to putative virulence targets (*Cesari et al., 2014*; *Wu et al., 2015*). Such domains can be integrated at different positions either before, in-between or after the standard NLR regions and are increasingly identified in NLRs of both model and crop plants (*Cesari et al., 2014*). How these unusual integrated domains function in the direct molecular recognition of effectors, and how this results in initiation of immune signalling, are emerging as fundamental questions in plant NLR biology.

To date, ~100 NLRs in rice have been described to confer resistance to strains of *M. oryzae*, and 23 of these have been cloned (*Liu et al., 2014*). Identification of the pathogen effectors (also known as AVRs [avirulence proteins]) that are recognised by these NLRs has lagged behind and only six have been cloned to date, AVR-Pia (*Yoshida et al., 2009*), AVR-Pita (*Orbach et al., 2000*), AVR-Pik (*Yoshida et al., 2009*), AVR-Pii (*Yoshida et al., 2009*), AVR-Piz-t (*Li et al., 2009*) and AVR1-CO39 (*Ribot et al., 2013*). AVR-Pia and AVR1-CO39 are recognised by the RGA4/RGA5 NLR pair (*Okuyama et al., 2011*; *Cesari et al., 2013*) through direct binding to a Heavy-Metal Associated domain (HMA, also known as RATX1) integrated into RGA5 after the LRR (*Cesari et al., 2013*). RGA4/RGA5 physically interact to prevent cell death mediated by RGA4 in the absence of AVR-Pia; the presence of the effector relieves this suppression (*Cesari et al., 2014*). Intriguingly, the NLR pair Pik-1/Pik-2, which

recognises AVR-Pik (*Figure 1*) (*Ashikawa et al., 2008*; *Yoshida et al., 2009*), also binds the effector via an HMA domain but this domain is integrated between the CC and NB-ARC regions of Pik-1 (*Figure 1B*). The integrated HMA domains of RGA5 and Pik-1 appear to have evolved from a family of rice proteins that only contain the HMA domain (*Cesari et al., 2013*, *2014*; *Wu et al., 2015*). Interestingly, the rice protein Pi21, a disease susceptibility factor, contains an HMA domain that is not part of an NLR protein (*Fukuoka et al., 2009*).

Both AVR-Pik (*Figure 1A*) and the HMA region of Pik-1 exhibit nucleotide polymorphisms between pathogen isolates and rice cultivars that result in changes at the amino acid level (*Yoshida et al., 2009*; *Kanzaki et al., 2012*; *Wu et al., 2014*; *Zhai et al., 2014*). These changes are most likely associated with co-evolutionary dynamics between *M. oryzae* and rice, predicted to play out at the molecular level via direct protein:protein interactions (*Kanzaki et al., 2012*). The interaction of AVR-Pik allele AVR-PikD with the Pik-1 NLR Pikp-1 is thought to be the oldest in co-evolutionary time (*Kanzaki et al., 2012*). Cultivars of rice containing the Pikp allele are resistant to *M. oryzae* isolates expressing AVR-PikD, but are susceptible to pathogen isolates expressing other AVR-Pik alleles (*Kanzaki et al., 2012*).

While the structural basis of function and recognition of plant pathogen effectors has advanced in recent years (*Wirthmueller et al., 2013*; *Williams et al., 2014*), only a few studies have focused on the *M. oryzae*/rice system. For example, the only structure known for a *M. oryzae* effector is that of AVR-Piz-t (*Zhang et al., 2013*), which adopts a six-stranded β-sandwich structure and contains a single disulphide bond. To date, there is no available structural information on domains from rice NLRs, and no structural data from any system showing how plant pathogen effectors are directly recognised at the molecular level by an NLR.

To better understand the mechanisms of direct recognition of effectors by NLRs, we have investigated the interaction between the *M. oryzae* effector AVR-Pik and the rice NLR Pikp-1. We determined the affinity of interaction of AVR-PikD to the HMA domain of the rice NLR Pikp-1 (Pikp-HMA) in vitro, and compared the relative binding of AVR-Pik alleles AVR-PikE, AVR-PikA and AVR-PikC to this HMA. The crystal structure of AVR-PikD bound to Pikp-HMA was determined and this guided mutagenesis of the effector, targeting residues at the interface with Pikp-HMA. The binding of these mutants to the Pikp-HMA was tested in yeast and in vitro. We also used a combination of AVR-Pik alleles and AVR-PikD mutants in both the host rice and heterologous *Nicotiana benthamiana* systems to probe the degree to which AVR-NLR interactions mediate immunity-related readouts. Long after Harold Henry Flor proposed the gene-for-gene hypothesis of host–parasite interactions (*Flor, 1955*, *1971*), our study establishes the structural basis of direct recognition of a pathogen effector by a plant NLR.

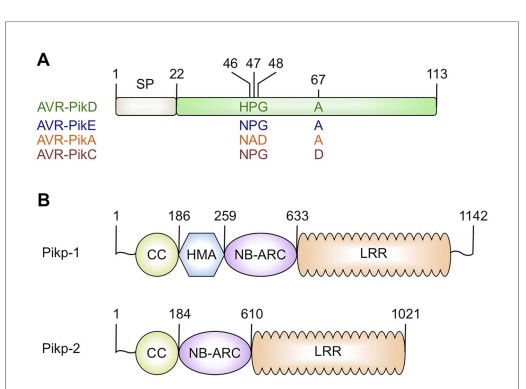

**Figure 1**. Schematic representations of (**A**) *Magnaporthe oryzae* AVR-Pik effector alleles with position of polymorphic residues shown, the effector domain is shown in green with the signal peptide (SP) in grey (amino acids are denoted by their single letter codes), (**B**) Rice Pik resistance proteins, highlighting the position of integrated HMA domain in the classical plant NLR architecture of Pik-1 (CC = coiled coil, HMA—Heavy Metal Associated domain, NB-ARC = Nucleotide-binding Apaf-1, R-protein, CED4-shared domain, LRR = Leucine Rich Repeat domain), domain boundaries are numbered, based on the Pikp sequences.

## Results

### The rice NLR Pikp-HMA domain selectively interacts with *M. oryzae* effector AVR-PikD in yeast and in vitro

Previously, the full-length and CC domain (containing the HMA) of Pikp-1 have been shown to interact with AVR-PikD in yeast-2-hybrid (Y2H) assays (*Kanzaki et al., 2012*; *Wu et al., 2014*; *Zhai et al., 2014*). Here we show that the Pikp-HMA domain alone selectively interacts with the AVR-Pik allele AVR-PikD in yeast (*Figure 2A*, *Figure 2—figure supplement 1*, *Table 1*). Weak interaction was also observed with AVR-PikE (as evidenced by limited growth on the

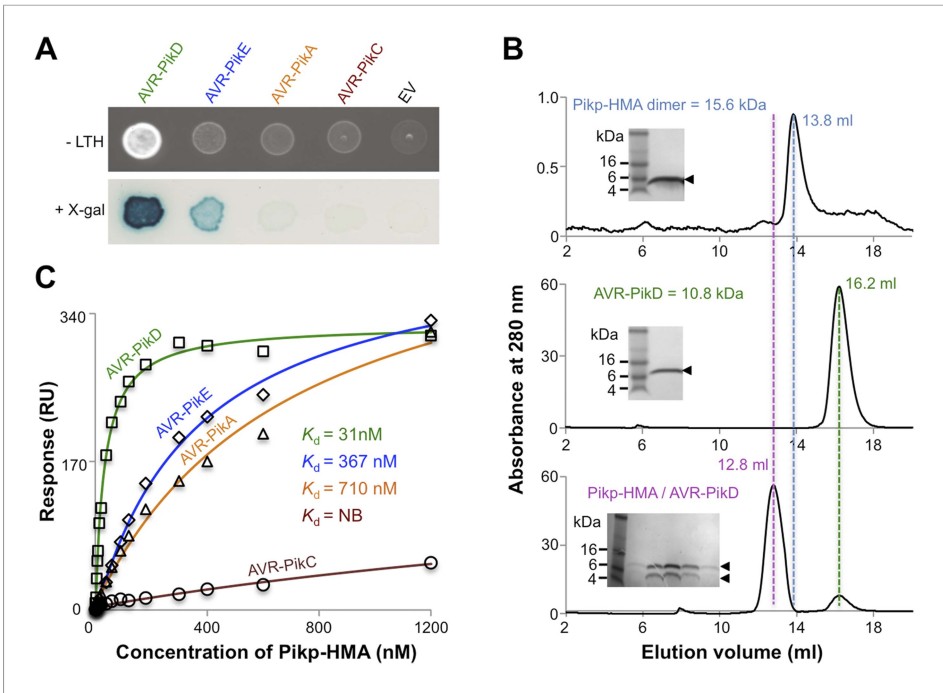

**Figure 2**. AVR-Pik effector alleles interact with the Pikp-HMA domain with different affinities. (**A**) Y2H assays showing the binding of effector alleles to the Pikp-HMA using two read-outs, growth on–Leu-Trp-His+3AT (-LTH) plates and the X-gal assay. (**B**) Analytical Gel Filtration traces depicting the retention volume of Pikp-HMA, AVR-PikD and the complex, with SDS-PAGE gels of relevant fractions (similar results were obtained for AVR-PikE and AVR-PikA, but AVR-PikC did not bind [*Figure 2—figure supplement 3*]). (**C**) Binding curves derived from Surface Plasmon Resonance multi-cycle kinetics data for Pikp-HMA binding to AVR-Pik alleles, $K_d$ values are shown (NB = No Binding). The sensorgrams of the data used to derive these curves are shown in *Figure 2—figure supplement 4*.

The following figure supplements are available for figure 2:

**Figure supplement 1**. Confirmation of protein expression in yeast.

**Figure supplement 2**. SDS-PAGE gels of purified proteins.

**Figure supplement 3**. Analytical gel filtration.

**Figure supplement 4**. SPR sensorgrams.

selective–LTH plate and some blue colouration in the X-gal assay), but not AVR-PikA or AVR-PikC, which is consistent with previous experiments (*Kanzaki et al., 2012*). Following expression, purification and verification of the proteins by intact mass spectrometry ('Materials and methods', *Figure 2—figure supplement 2*, *Table 2*), we showed that AVR-PikD and Pikp-HMA form a stable complex in vitro that can be purified by analytical gel filtration (*Figure 2B*). Using this qualitative assay, we also found that Pikp-HMA can form a complex with AVR-PikE and AVR-PikA, but not AVR-PikC (*Figure 2—figure supplement 3*).

Next we used Surface Plasmon Resonance (SPR) to determine the binding affinity of AVR-PikD to Pikp-HMA. The purified effector, with a non-cleavable 6xHis tag at the C-terminus, was immobilised on a $Ni^{2+}$-NTA chip; Pikp-HMA was used as the analyte. Using a multi-cycle kinetics approach, we found that Pikp-HMA bound to immobilised AVR-PikD with a $K_d$ of 31 ± 2 nM (*Figure 2C*, *Figure 2—figure supplement 4A*, *Table 1*). SPR studies were expanded to include AVR-PikE, AVR-PikA and AVR-PikC (*Figure 2C*, *Figure 2—figure supplement 4*, *Table 1*). For AVR-PikE, even though this was not fully saturatable under the conditions of our assay, we obtained an apparent $K_d$ of 367 ± 41 nM, a greater than 10-fold weaker binding compared to AVR-PikD (*Figure 2C*). For AVR-PikA we could determine an

**Table 1**. Summary Table showing the outcomes of in vitro and *in planta* assays used to investigate the interactions and responses of AVR-Pik effectors with Pikp-dependent readouts

| | AVR-PikD | AVR-PikE | AVR-PikA | AVR-PikC | AVR-PikD[His46Glu] | AVR-PikD[Ile49Glu] | AVR-PikD[Arg64Ala] | AVR-PikD[Asp66Arg] | AVR-PikD[Ala67Asp] | AVR-PikD[Pro47Ala/Gly48Asp] |
|---|---|---|---|---|---|---|---|---|---|---|
| Interaction with Pikp-HMA in Y2H | +++ | + | – | – | – | +++ | – | – | ++ | + |
| Interaction with Pikp-HMA in SPR | +++ | ++ | + | – | – | ++ | – | – | + | ++ |
| Recognition in Pikp⁺ rice plants | +++ | + | (–) | (–) | – | + | – | – | +++ | +++ |
| CD response in *Nicotiana benthamiana* | +++ | – | – | – | – | ++ | – | – | – | +++ |

Y2H = yeast-2-hybrid, SPR = Surface Plasmon Resonance, Pikp⁺ = rice cv. K60, CD = cell death. Parentheses depict results from (**Kanzaki et al., 2012**).

apparent $K_d$ of 710 ± 111 nM (also not saturable in the assay). We detected essentially no binding for AVR-PikC to Pikp-HMA in this assay (*Figure 2C*). This is consistent with the Y2H data and also correlates with the published recognition specificity *in planta*, although *M. oryzae* isolates expressing AVR-PikE were reported to not be recognised by cultivars of rice expressing Pikp (*Kanzaki et al., 2012*).

## Crystal structure of the Pikp-HMA/AVR-PikD complex

Although we were able to express and purify AVR-PikD from *Escherichia coli*, we were unable to obtain crystals of this protein for structure determination by X-ray crystallography. However, following a co-expression strategy with 6xHis tagged Pikp-HMA and untagged AVR-PikD (see 'Materials and

**Table 2**. Intact masses for proteins expressed and purified in this study

| Protein | Vector | Molecular Mass (Da) | | |
|---|---|---|---|---|
| | | Calculated | Observed | Δ |
| Pikp-HMA | pOPINS3C* | 7805.23 | 7804.97 | −0.26 |
| AVR-PikD | pOPINS3C* | 10,835.31 | 10,832.95 | −2.36§ |
| AVR-PikD | pOPINA† | 10,812.33 | 10,809.99 | −2.34 |
| AVR-PikD | pOPINE‡ | 11,786.33 | 11,784.16 | −2.17 |
| AVR-PikE | pOPINS3C* | 10,812.27 | 10,809.91 | −2.36 |
| AVR-PikE | pOPINE‡ | 11,763.29 | 11,760.96 | −2.33 |
| AVR-PikA | pOPINS3C* | 10,844.27 | 10,841.80 | −2.47 |
| AVR-PikA | pOPINE‡ | 11,795.29 | 11,793.01 | −2.28 |
| AVR-PikC | pOPINS3C* | 10,856.28 | 10,853.72 | −2.56 |
| AVR-PikC | pOPINE‡ | 11,807.30 | 11,804.97 | −2.33 |
| AVR-PikD[His46Glu] | pOPINE‡ | 11,778.30 | 11,776.07 | −2.23 |
| AVR-PikD[Ile49Glu] | pOPINE‡ | 11,802.28 | 11,800.04 | −2.24 |
| AVR-PikD[Arg64Ala] | pOPINE‡ | 11,701.22 | 11,698.94 | −2.28 |
| AVR-PikD[Asp66Arg] | pOPINE‡ | 11,827.43 | 11,825.31 | −2.12 |
| AVR-PikD[Ala67Asp] | pOPINE‡ | 11,830.34 | 11,828.20 | −2.14 |
| AVR-PikD[Pro47Ala, Gly48Asp] | pOPINE‡ | 11,818.32 | 11,816.20 | −2.12 |

*Non-native residues remaining after 3C cleavage: N-terminal Gly–Pro.
†Non-native residues remaining: N-terminal Met.
‡Non-native residues remaining after 3C cleavage: N-terminal Gly–Pro; C-terminal Lys-His-His-His-His-His-His.
§The measured mass of each AVR-Pik protein should be 2.0156 Da (2 × 1.0078) less than its calculated mass due to formation of the di-sulphide bond.

methods'), we obtained crystals of this complex in multiple conditions. Optimisation of one of these conditions (*Figure 3—figure supplement 1*) produced crystals diffracting X-rays to 1.6 Å resolution. The structure of the Pikp-HMA/AVR-PikD complex was solved using molecular replacement to position a Pikp-HMA dimer (see below) in the asymmetric unit, followed by automated rebuilding with the sequence of both proteins supplied. This was sufficient to produce an initial model containing both Pikp-HMA and AVR-PikD that could be used to complete structure determination (see 'Materials and methods', X-ray Data Collection and Refinement statistics are given in *Table 3*).

The structure of the Pikp-HMA/AVR-PikD complex reveals an intimate interface formed between these proteins that buries 18.7% of the effector's solvent accessible surface area (1031.0 Å$^2$, *Figure 3A,B*). The majority of the interaction is formed with a monomer of Pikp-HMA, with 87.5% of the effector's buried surface area (902.2 Å$^2$) and nine residues contributing hydrogen bond and/or salt bridge interactions. This suggests that the AVR-PikD/Pikp-HMA monomer interaction most likely represents the biologically significant interface. No hydrogen bonds or salt bridge interactions are formed with the second monomer of the Pikp-HMA dimer. Further, due to steric clash that would occur, it is not possible for an AVR-PikD/Pikp-HMA heterotetramer (2:2 complex) to assemble. All interface analysis was performed using PDBePISA (*Krissinel and Henrick, 2007*).

## Structure of Pikp-HMA in the Pikp-HMA/AVR-PikD complex

Each of the Pikp-HMA monomers adopts the HMA-domain fold (Pfam: PF00403), comprising a four-stranded antiparallel β-sheet and two α-helices packed in an α/β sandwich. The closest structural homologue of Pikp-HMA (defined by PDBeFold [*Krissinel and Henrick, 2004*]) is the HMA domain of yeast protein Ccc2A (*Banci et al., 2001*), overlaying with an r.m.s.d. of 1.58 Å over 72 residues (*Figure 3C*). Typically, HMA domains bind heavy metals, or lighter cations such as Cu$^{1+}$ or Zn$^{2+}$, via two conserved Cys residues and are involved in metal transport or detoxification pathways (*Bull and Cox, 1994*). Interestingly, these Cys residues are not conserved in Pik-1 HMA domains, including Pikp-1. Hence, the Pikp-HMA structure does not contain a metal ion, and the loop between β$_1$ and α$_1$, which usually contains the metal-chelating Cys residues, is disordered. Further, this loop is positioned away from the interface with the effector (*Figure 3A*, *Figure 3—figure supplement 2*) and does not contribute to complex formation.

We were also able to obtain the crystal structure of Pikp-HMA in the absence of AVR-PikD (see 'Materials and methods', *Figure 3—figure supplements 1, 2A*). The structure of the Pikp-HMA dimer in isolation is essentially identical to that found in the complex (r.m.s.d. 0.67 Å over 69 residues, for the monomer bound to AVR-PikD), with the exceptions of a minor shift in the loop spanning residues Val222—Lys228 and the N-terminal four residues (*Figure 3—figure supplement 2C*).

## Structure of AVR-PikD in the Pikp-HMA/AVR-PikD complex

AVR-PikD adopts a six-stranded β-sandwich structure, stabilised by a di-sulphide bond between Cys54 and Cys70. The effector contains an N-terminal extension, comprising residues Arg31 to Pro52, prior to the start of this fold (*Figure 3A,D*). The extension is anchored to the β-sandwich at each end via a salt–bridge interaction involving the side chains of Asp45 and Arg110 and hydrogen bonds between both the main chain carbonyl of Arg39 and Glu38$^{O\epsilon1}$ and Arg64$^{N\eta1}$.

Database searches using PDBeFold reveals that a close structural homologue of AVR-PikD is AVR-Piz-t (*Zhang et al., 2013*), another *M. oryzae* effector protein, despite there being essentially no sequence identity between these proteins (r.m.s.d. = 2.33 Å over 58 aligned residues, *Figure 3D*). This suggests that sequence divergent translocated effectors of *M. oryzae* may share a conserved structural scaffold, despite very different sequences, which has striking parallels to RXLR-type effectors of plant pathogenic oomycetes (*Boutemy et al., 2011*; *Win et al., 2012*). Further, structural homology is also observed to ToxB, a protein toxin from *Pyrenophora tritici-repentis*, the causative agent of tan spot in wheat (*Nyarko et al., 2014*). In each case, the identified structural homology only extends to the β-sandwich fold and the N-terminal extension of AVR-PikD appears to be unique. This raises the interesting possibility that candidate effectors from distant fungi could be identified by structure-guided sequence similarity searches.

## Binding interfaces in the Pikp-HMA/AVR-PikD complex

Three primary sites of interaction are apparent between Pikp-HMA and AVR-PikD. The first is dominated by main-chain hydrogen bonding between the C-terminal β-stand of Pikp-HMA and β$_3$ of AVR-PikD, which results in formation of a continuous antiparallel β-sheet comprising the four β-strands

**Table 3**. X-ray data collection and refinement statistics

| | Pikp-HMA | | Pikp-HMA/AVR-PikD |
|---|---|---|---|
| | **Native** | **Iodide** | |
| Data collection | | | |
| Wavelength (Å) | 1.20 | 2.00 | 0.90 |
| Space group | $P6_522$ | $P6_522$ | $P4_12_12$ |
| Cell dimensions | | | |
| $a$, $b$, $c$ (Å) | 54.65, 54.65, 235.22 | 54.73, 54.73, 235.80 | 118.41, 118.41, 35.81 |
| $\alpha$, $\beta$, $\gamma$, (°) | 90.00, 90.00, 120.00 | 90.00, 90.00, 120.00 | 90.00, 90.00, 90.00 |
| Resolution (Å)* | 47.33–2.10 (2.15–2.10) | 117.90–2.80 (2.87–2.80) | 39.47–1.60 (1.64–1.60) |
| $R_{merge}$ (%) | 8.4 (117.6) | 8.7 (45.8) | 4.7 (65.1) |
| $I/\sigma I$ | 32.3 (4.6) | 34.7 (7.3) | 32.3 (4.7) |
| Completeness (%) | | | |
| Overall | 100 (99.9) | 99.9 (98.9) | 100 (100) |
| Anomalous | | 99.9 (99.4) | |
| Redundancy | | | |
| Overall | 45 (46.8) | 32.8 (24.4) | 17.7 (17.4) |
| Anomalous | | 19.4 (13.3) | |
| $CC^{(1/2)}$ (%) | 100 (94.0) | 100 (98.0) | 100 (92.8) |
| Refinement and model | | | |
| Resolution (Å) | 47.33–2.10 (2.15–2.10) | | 39.47–1.60 (1.64–1.60) |
| Reflections | 12356 (861) | | 32549 (2379) |
| $R_{work}/R_{free}$ (%) | 20.2/22.9 (20.6/19.6) | | 17.8/20.5 (19.6/24.7) |
| No. atoms | | | |
| Protein | 1063 | | 1762 |
| Water | 44 | | 138 |
| B-factors (Å²) | | | |
| Protein | 29.96 | | 23.38 |
| Water | 57.31 | | 34.07 |
| R.m.s deviations | | | |
| Bond lengths (Å) | 0.013 | | 0.016 |
| Bond angles (°) | 1.57 | | 1.79 |
| Ramachandran plot (%)† | | | |
| Favoured | 97.1 | | 98.7 |
| Allowed | 2.9 | | 1.3 |
| Outliers | 0 | | 0 |
| MolProbity Score | 1.48 (98th percentile) | | 1.21 (98th percentile) |

*The highest resolution shell is shown in parentheses.
†As calculated by MolProbity.

of Pikp-HMA and β[3–5] of AVR-PikD. The second involves the side chain of Pikp-HMA[Asp224], which forms a salt–bridge interaction with the side chain of AVR-PikD[Arg64], and is also held in place by a hydrogen bond of its main chain NH group to the side chain of AVR-PikD[Asp66] (*Figure 3A*). The third interaction site centres on AVR-PikD[His46], although has contributions from residues Asn42—Ile49. This region forms part of the N-terminal extension and includes the polymorphic AVR-Pik residues 46, 47 and 48 (His46, Pro47 and Gly48 in AVR-PikD, *Figures 1A and 3A*). AVR-PikD[His46]

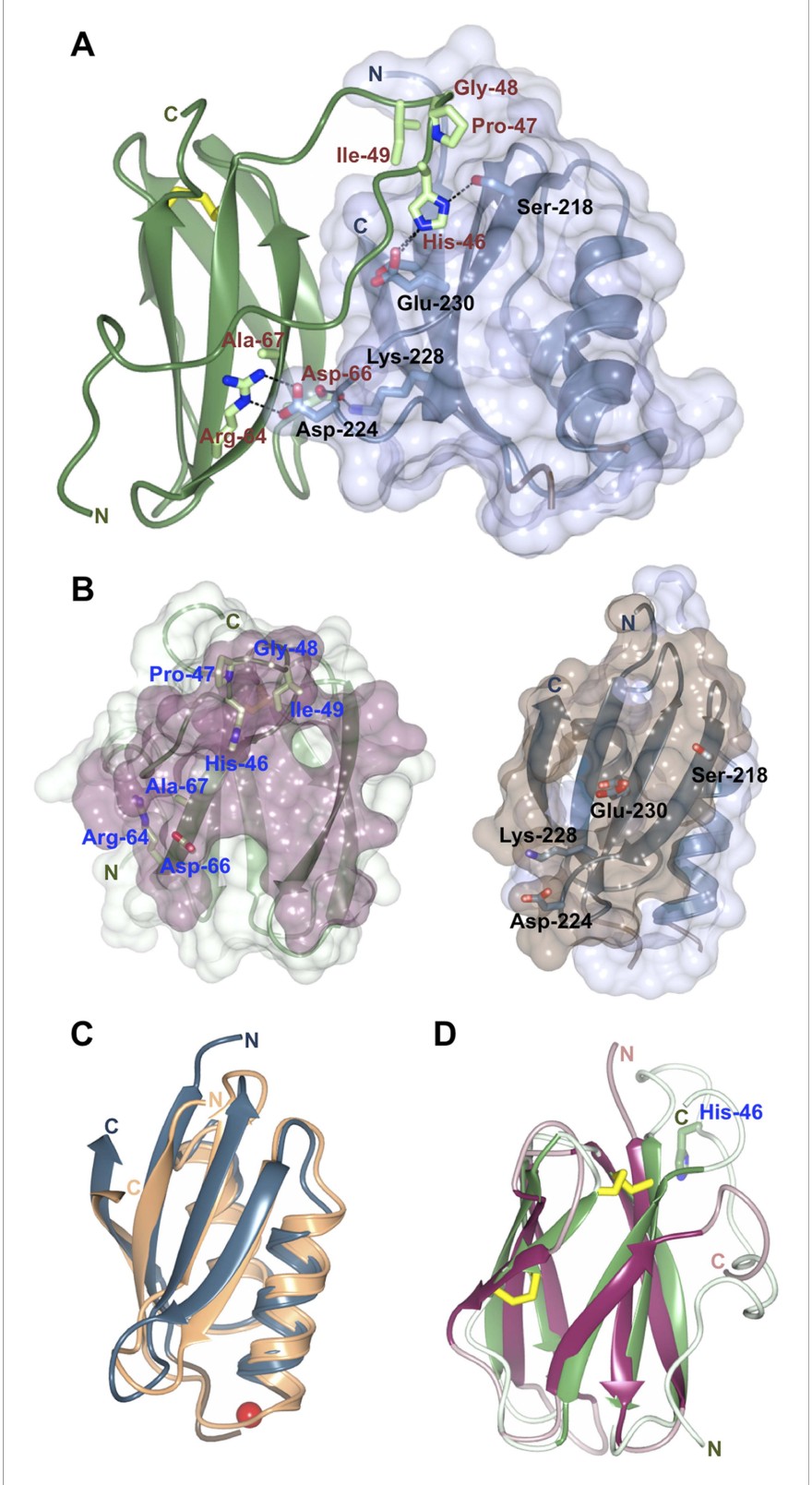

**Figure 3**. Structure of the AVR-PikD/Pikp-HMA complex. (**A**) Schematic representation of the AVR-PikD/Pikp-HMA (monomer), highlighting interfacing residues. The effector is shown in green cartoon, with side chains as sticks and green carbon atoms (no surface). The Pikp-HMA is shown in blue cartoon, with side chains as sticks and blue carbon atoms; the molecular surface of this protein is also depicted. Effector residues selected for mutation are labelled, as are important

*Figure 3. continued on next page*

*Figure 3. Continued*

interface residues of Pikp-HMA discussed in the text. Hydrogen bonds/salt-bridges are shown as dashed lines and the disulphide bond as yellow bars. (**B**) Buried surface areas of AVR-PikD (left, purple) and Pikp-HMA (right, brown) separated and shown from the perspective of the partner molecule. Cartoon and amino acid side chains shown are as for panel (**A**). (**C**) Comparison of the Pikp-HMA (monomer, blue) with yeast Ccc2A (wheat) showing the conservation of the HMA fold. The copper ion bound to Ccc2a is shown as a red sphere. (**D**) Comparison of AVR-PikD (green) and AVR-Piz-t (pink) structures showing the conservation of the β-sandwich structure, and the N-terminal extension of AVR-PikD.

The following figure supplements are available for figure 3:

**Figure supplement 1**. Sample preparation for x-ray data collection.

**Figure supplement 2**. The structure of the Pikp-HMA dimer is conserved when bound to AVR-PikD.

**Figure supplement 3**. Polymorphic residue AVR-PikD$^{His46}$ is bound within a pocket on the Pikp-HMA surface.

**Figure supplement 4**. Amino acid sequence alignment of AVR-Pik alleles and Pik-HMA domains.

is bound in a pocket on Pikp-HMA via hydrogen bonds/salt bridge interactions between AVR-PikD$^{His46:Nδ1}$/Pikp-HMA$^{Ser218:Oγ}$ and AVR-PikD$^{His46:Nε2}$/Pikp-HMA$^{Glu230:Oε1}$; also, Pikp-HMA$^{Val232}$ packs on top of the AVR-PikD$^{His46}$ ring and contributes hydrophobic/van der Waals interactions (*Figure 3—figure supplement 3*). Finally, it is worth noting that there is an extensive network of buried solvent-mediated contacts between Pikp-HMA and AVR-PikD.

## Structure-based mutations in AVR-PikD perturb binding to Pikp-HMA in yeast and in vitro

Based on the Pikp-HMA/AVR-PikD structure, we designed four mutations in AVR-PikD predicted to perturb complex formation through generating steric clashes/introducing charged residues, or removing a salt–bridge interaction (His46Glu, Ile49Glu, Asp66Arg and Arg64Ala), and two mutants to mimic other AVR-Pik alleles, but retain His46 (Ala67Asp [based on AVR-PikC], Pro47Ala/Gly48Asp [based on AVR-PikA]), *Figure 3—figure supplement 4A*.

First, we screened these mutants for interaction with Pikp-HMA in the Y2H assay. We found that AVR-PikD$^{His46Glu}$, AVR-PikD$^{Arg64Ala}$ and AVR-PikD$^{Asp66Arg}$ prevent the interaction (as observed on the -LTH selective growth plate and in the X-gal assay (*Figure 4A*, *Figure 4—figure supplement 1*, *Table 1*). However, AVR-PikD$^{Ile49Glu}$ maintains an interaction and AVR-PikD$^{Ala67Asp}$ and AVR-PikD$^{Pro47Ala/Gly48Asp}$ showed intermediate binding (weak interaction on -LTH selective growth plate and in the X-gal assay [*Figure 4A*]).

Next, we expressed and purified each of these AVR-PikD mutants (as for wild-type and with C-terminal non-cleavable 6xHis tag) and confirmed their identity by intact mass spectrometry (*Figure 4—figure supplement 2A*, *Table 2*). We then used SPR to determine the binding affinities between these mutants and the Pikp-HMA using a single-cycle kinetics approach ([*Karlsson et al., 2006*] *Figure 4B*, *Figure 4—figure supplement 2B*, *Table 1*), having confirmed a similar affinity of Pikp-HMA for AVR-PikD ($K_d$ = 29 ± 3.5 nM) using this approach. Consistent with the Y2H results, we could not measure any meaningful interaction of AVR-PikD$^{His46Glu}$, AVR-PikD$^{Arg64Ala}$ and AVR-PikD$^{Asp66Arg}$ with Pikp-HMA (*Figure 4B*). For AVR-PikD$^{Ile49Glu}$ and AVR-PikD$^{Pro47Ala/Gly48Asp}$ we were able to determine $K_d$s of interaction of 99 ± 18 nM and 83 ± 16 nM respectively (*Figure 4B*, *Figure 4—figure supplement 2B*). AVR-PikD$^{Ala67Asp}$ showed a weaker response but we were unable to obtain a reliable $K_d$ at the concentrations of Pikp-HMA used. AVR-PikD$^{Ile49Glu}$, AVR-PikD$^{Ala67Asp}$ and AVR-PikD$^{Pro47Ala/Gly48Asp}$ all interacted in the Y2H assay, with the latter two showing qualitatively weaker binding.

## Structure-based mutations in AVR-PikD prevent response in rice when delivered by *M. oryzae*

To test the effects of the AVR-PikD mutations on pathogen virulence on rice plants expressing the Pikp gene, we transformed *M. oryzae* isolate Sasa2 with constructs encoding each of the six

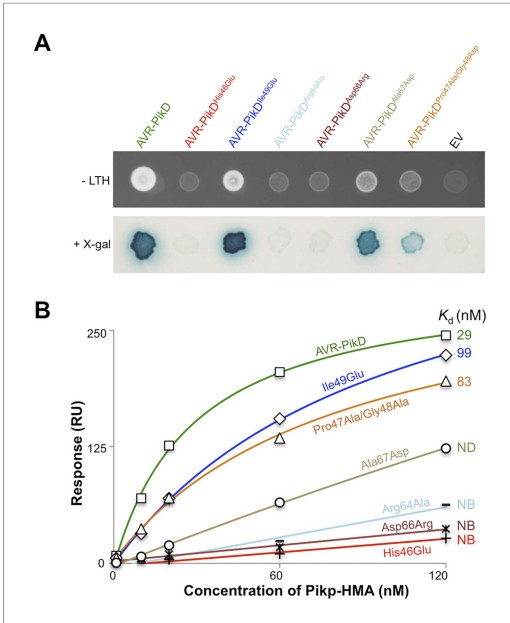

**Figure 4**. Structure-based mutagenesis at the Pikp-HMA/AVR-PikD interface perturbs protein interactions in yeast and in vitro. (**A**) Y2H assays showing the binding of AVR-PikD mutants to Pikp-HMA using two read-outs, growth on–Leu-Trp-His+3AT (-LTH) plates and the X-gal assay. (**B**) Binding curves derived from Surface Plasmon Resonance single-cycle kinetics data for Pikp-HMA binding to AVR-PikD and AVR-PikD mutants, $K_d$ values are shown where determined (ND = Not Determined, NB = No Binding). The sensorgrams of the data used to derive these curves are shown in *Figure 4—figure supplement 2B*.

The following figure supplements are available for figure 4:

**Figure supplement 1**. Confirmation of protein expression in yeast.

**Figure supplement 2**. SDS-PAGE of AVR-PikD mutant proteins and SPR sensorgrams.

mutants above, with expression driven by the native AVR-PikD promoter. AVR-PikE was included in these experiments as it represents a naturally occurring point mutant at the important position 46 (His46Asn). Each of the transformed *M. oryzae* lines were spot inoculated (*Kanzaki et al., 2002*) onto leaf blades of rice cultivars Nipponbare (Pik⁻, lacks known Pik alleles) and K60 (which contains Pikp). The Nipponbare cultivar was susceptible to all of the *M. oryzae* lines, including Sasa2 wild type and empty vector control, as shown by the development of lesions around the inoculation sites (*Figure 5*). As expected, the K60 (Pikp) cultivar is resistant to the *M. oryzae* Sasa2 line expressing AVR-PikD (*Kanzaki et al., 2012*). We observed that the K60 (Pikp) cultivar showed an intermediate phenotype between susceptible and resistant to *M. oryzae* Sasa2 lines expressing AVR-PikE. For the mutants, we found that the K60 (Pikp) cultivar was susceptible to *M. oryzae* Sasa2 lines expressing AVR-PikD^His46Glu, AVR-PikD^Arg64Ala and AVR-PikD^Asp66Arg, but resistant to those expressing AVR-PikD^Ala67Asp and AVR-PikD^Pro47Ala/Gly48Asp (*Figure 5*, *Table 1*). As for AVR-PikE, K60 (Pikp) shows an intermediate phenotype to Sasa2 lines expressing AVR-PikD^Ile49Glu (*Figure 5*). There is a correlation between AVR-PikD mutants that display the tightest binding affinities in vitro, and interact in the Y2H assay, with resistance in rice when delivered by *M. oryzae* (partial phenotype in the case of AVR-PikD^Ile49Glu). Strains expressing AVR-PikD mutants that do not interact in vitro or in yeast (AVR-PikD^His46Glu, AVR-PikD^Arg64Ala and AVR-PikD^Asp66Arg) are fully susceptible in rice. Expression of AVR-PikD and mutants in the transgenic *M. oryzae* during infection was confirmed by RT-PCR (*Figure 5—figure supplement 1*).

## Pikp/AVR-PikD co-expression in *N. benthamiana* elicits a cell death response

To further investigate the link between recognition of AVR-PikD by Pikp-1/Pikp-2 and immunity-related signalling, we established a transient expression assay in *N. benthamiana* leaves using *Agrobacterium tumefaciens* to deliver these genes into plant cells (henceforth Agroinfiltration, *Figure 6A*). In this system we required the co-expression of Pikp-1, Pikp-2 and AVR-PikD to observe robust features of cell death, including necrotic tissue and accumulation of phenolic compounds that give rise to auto-fluorescence (*Figure 6B*) (*Bos et al., 2006*). We did not observe significant effects in *N. benthamiana* leaves following expression of the individual proteins or any combination of protein pairs (*Figure 6B*). Further, co-expression of AVR-PikE, AVR-PikA and AVR-PikC with Pikp-1 and Pikp-2 fails to elicit a cell death response (*Figure 6—figure supplements 1–4*). This demonstrates only the specific combination of Pikp-1, Pikp-2 and AVR-PikD results in a robust response and is consistent with the observed

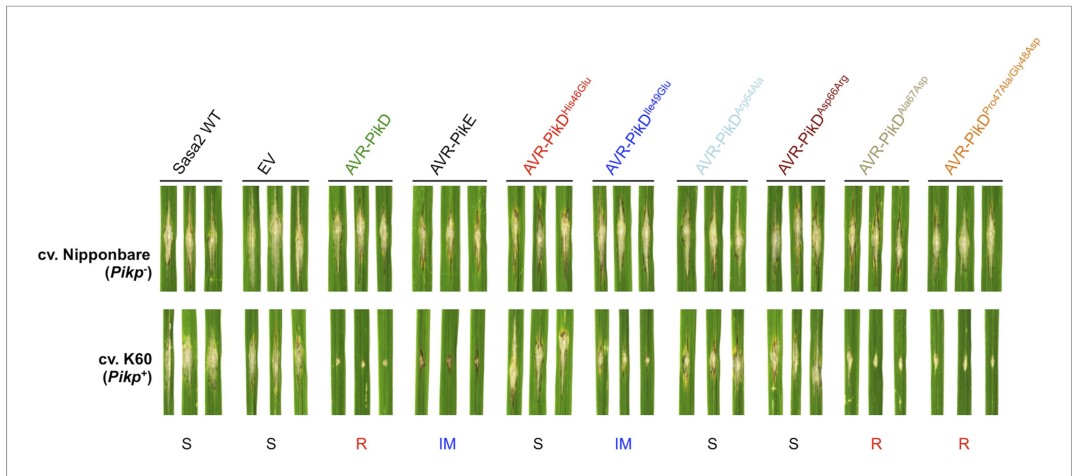

**Figure 5**. Structure-based mutagenesis at the Pikp-HMA/AVR-PikD interface leads to susceptibility in Pikp$^+$ rice plants. Rice plants Pik$^-$ (cv. Nipponbare) and Pikp$^+$ (cv. K60) were spot-inoculated with *M. oryzae* Sasa2 expressing AVR-PikD, AVR-PikE and AVR-PikD mutants. The combinations resulting in resistant (R), intermediate (IM) and susceptible (S) phenotype are labelled.

The following figure supplement is available for figure 5:

**Figure supplement 1**. RT-PCR.

pairings between rice cultivars with different Pik alleles and *M. oryzae* isolates with different AVR-Pik alleles (*Kanzaki et al., 2012*).

## Structure-based mutations in AVR-PikD prevent Pikp-mediated cell death in *N. benthamiana*

To further correlate the cell death response with direct protein:protein interaction, we co-expressed Pikp-1, Pikp-2 and the AVR-PikD mutants described above via Agroinfiltration in *N. benthamiana* leaves (*Figure 7—figure supplement 1*). Using this approach, we found that the AVR-PikD$^{His46Glu}$, AVR-PikD$^{Arg64Ala}$, AVR-PikD$^{Asp66Arg}$ and AVR-PikD$^{Ala67Asp}$ mutations do not elicit a response, but AVR-PikD$^{Ile49Glu}$ and AVR-PikD$^{Pro47Ala/Gly48Asp}$ still promote cell death (*Figure 7*, *Figure 7—figure supplement 2*, *Table 1*). Interestingly, while AVR-PikD$^{Ile49Glu}$ consistently generates a response in *N. benthamiana* and bound to Pikp-HMA in the SPR and Y2H assays, it displayed an intermediate phenotype in the *M. oryzae* inoculation assay. Further, while AVR-PikD$^{Ala67Asp}$ did not elicit a response in *N. benthamiana* and showed only weak interaction by SPR, it did bind to Pikp-HMA in the Y2H assay, and induced resistance in the *M. oryzae* inoculation assay. These results suggest that differences in binding affinities between effectors and Pikp-HMA in vitro can occasionally result in subtly different readouts in plants.

## Discussion

Understanding how plant NLRs function at the molecular level is critical for their effective deployment in agriculture. Despite being >20 years since cloning of the first plant NLRs, this is still lacking. While single NLRs can be sufficient to mediate recognition and initiate signalling by either direct (*Dodds and Rathjen, 2010*) or indirect binding (*Dangl and Jones, 2001*; *van der Hoorn and Kamoun, 2008*), the role of paired NLRs is emerging as a new paradigm for regulating immune responses in plants and mammals. In such cases, one NLR acts as a pathogen 'sensor', and can contain a specific domain that mediates this activity, the second acts as an inducer of signalling. Recent studies have addressed the importance of molecular interactions between classical domains in plant NLR pairs (*Cesari et al., 2014*; *Williams et al., 2014*). Here we focussed on dissecting the direct recognition of a rice blast pathogen effector by an unconventional integrated domain (*Cesari et al., 2014*; *Wu et al., 2015*) in a rice NLR, a critical event for the initiation of immune-related signalling.

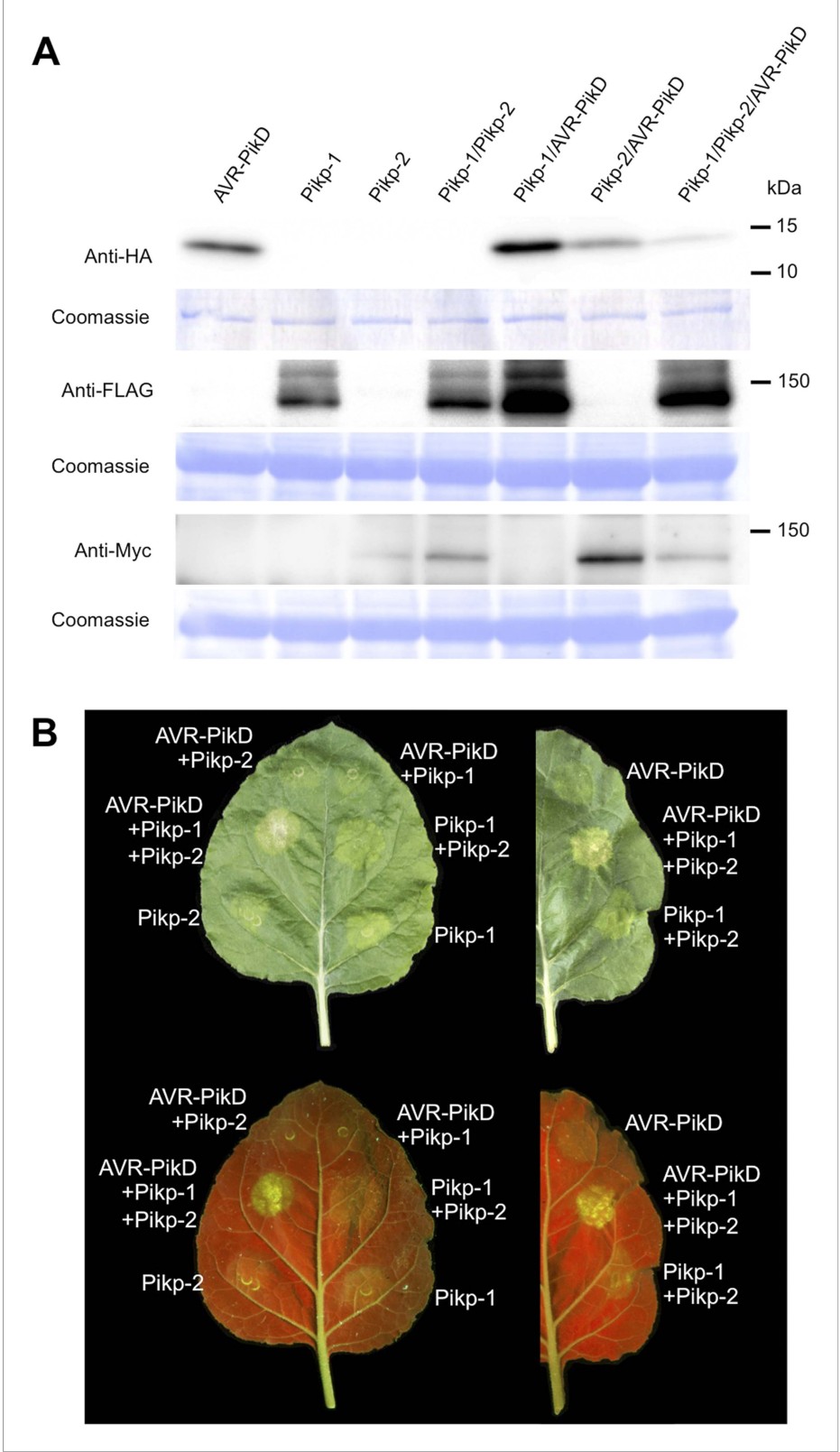

**Figure 6**. Pikp HR-like cell death in *Nicotiana benthamiana* requires co-delivery of AVR-PikD, Pikp-1 and Pikp-2. (**A**) Western blots showing expression of AVR-PikD(HA), Pikp-1(FLAG) and Pikp-2(Myc) in *N. benthamiana*. Blots were probed using the appropriate antibody for the tagged protein. (**B**) The Pikp HR-like cell death in *N. benthamiana* requires expression of Pikp-1, Pikp-2 and AVR-PikD together. Expression of individual proteins or co-expression of

*Figure 6. Continued*

any protein pair does not result in cell death. Images showing autofluorescence are horizontally flipped to present the same leaf orientation as white light images.

The following figure supplements are available for figure 6:

**Figure supplement 1**. Pikp HR-like cell death in *N. benthamiana* requires expression of Pikp-1, Pikp-2 and AVR-PikD specifically.

**Figure supplement 2**. Expression of AVR-Pik alleles alone, or in any combination with Pikp-1 or Pikp-2, does not result in HR-like cell death in *N. benthamiana* for (**A**) AVR-PikE, (**B**) AVR-PikA or (**C**) AVR-PikC.

**Figure supplement 3**. Example images used for scoring HR-like cell death (HR Index) in *N. benthamiana* on expression of Pikp-1, Pikp-2 and AVR-Pik alleles and AVR-PikD mutants.

**Figure supplement 4**. Box plots depicting HR Index for repeats of the assay shown in *Figure 6* and *Figure 6—figure supplement 1A*.

## Cell death signalling in the Pikp system requires Pikp-1, Pikp-2 and effector

In plants, paired NLRs such as rice RGA5/RGA4 and Arabidopsis RRS1/RPS4 (*Narusaka et al., 2009*), function through formation of homo- and hetero-protein complexes (*Cesari et al., 2014*; *Williams et al., 2014*). One member of the pair can constitutively activate an HR-like cell death on expression in plants (RGA4 and RPS4), and this activity is suppressed by the second (RGA5 and RRS1) through the formation of hetero-complexes (*Cesari et al., 2014*; *Williams et al., 2014*). This suppression is relieved by co-expression of the cognate effectors and can result in signalling-competent NLR homo-complexes (*Cesari et al., 2014*). In mammals, members of the NAIP (NLR) family act as sensors of pathogen signatures and associate with NLRC4 following perception to trigger signalling (*Kofoed and Vance, 2011*; *Zhao et al., 2011*). Although it seems unlikely that assemblies of homo- and hetero-NLR complexes in their suppressed and activated states are universally conserved in mammals and plants, it appears that oligomerisation plays a key role in modulating activity.

In contrast to RGA4 and RPS4, expression of Pikp-2 does not constitutively activate cell death in *Nicotiana*. In our assays, the HR-like response requires co-expression of Pikp-1 and AVR-PikD with Pikp-2, suggesting assembly of an active signalling complex requires all three proteins. Although the limits of our assays preclude a conclusive interpretation of this signalling complex, they do suggest that not all paired plant NLRs function within the confines of existing models. At present it is unknown whether or not Pikp-2 forms a heteromeric complex with Pikp-1 in an effector-dependent manner. Future work is required to dissect the underlying molecular interactions that promote signalling by the Pik NLRs.

## Polymorphic residue AVR-Pik[46] maps to the direct binding interface between Pikp-HMA and AVR-PikD

Phylogenetics suggests AVR-PikD is the ancestral AVR-Pik allele, and it is the only natural variant with a His at position 46 (*Kanzaki et al., 2012*). In the Pikp-HMA/AVR-PikD complex, the AVR-PikD[His46] side chain is buried within a pocket on the Pikp-HMA surface that contributes hydrogen bonds/salt bridge interactions (*Figure 3A*, *Figure 3—figure supplement 3*). The AVR-PikD[His46Glu] mutation prevents interaction with Pikp-HMA in vitro and in yeast, and response *in planta* either when delivered by *M. oryzae* into rice or on co-expression in *N. benthamiana*.

This data supports AVR-Pik[46] as a key site for recognition specificity, and that following introduction of Pikp into cultivated rice, *M. oryzae* evolved to evade recognition by mutating this residue. Interestingly, the only natural variant found at this position is a somewhat conservative His to Asn change (giving rise to AVR-PikE). Conceptually, this residue could be accommodated at the Pikp-HMA/AVR-PikD interface without generating significant steric clashes, but the interactions formed at this site (e.g., hydrogen bonding pattern) will be fundamentally different. This single

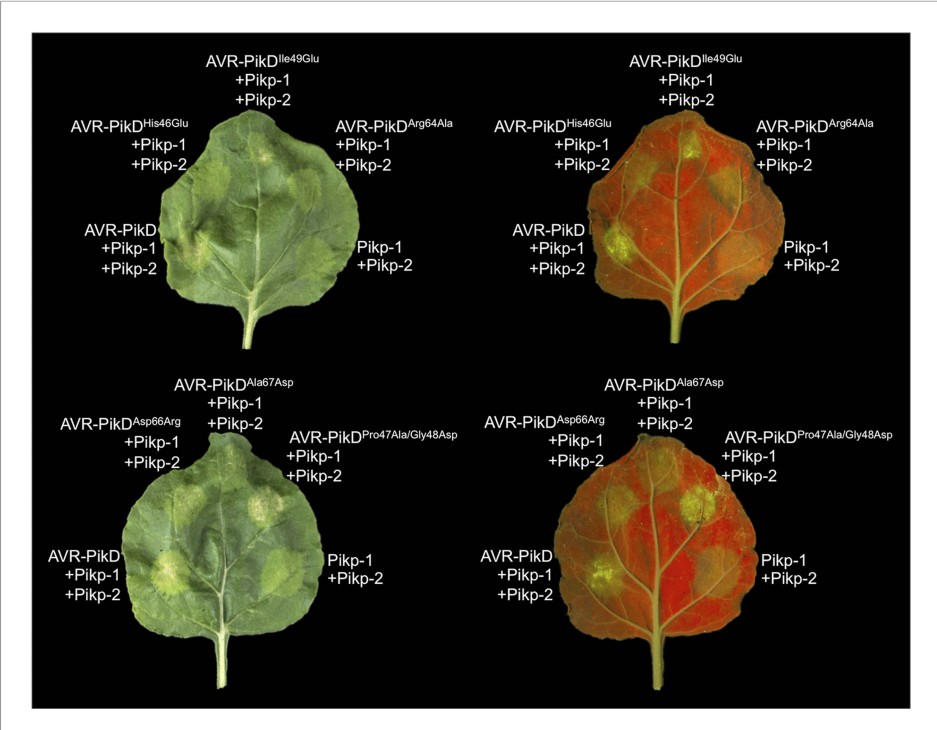

**Figure 7**. Structure based mutations at the AVR-PikD/Pikp-HMA interface leads to loss of HR-like cell death in *N. benthamiana*. Co-infiltration of Pikp-1 and Pikp-2 with AVR-PikD mutants His46Glu, Arg64Ala, Asp66Arg and Ala67Asp leads to loss of recognition and signalling in *N. benthamiana*. AVR-PikD$^{Ile49Glu}$ and AVR-PikD$^{Pro47Ala/Gly48Asp}$ retain recognition and signalling. Each infiltration site includes Pikp-1 and Pikp-2 with the AVR-PikD mutant indicated. Pikp-1 and Pikp-2 alone (EV) and with AVR-PikD are included as controls. Images showing autofluorescence are horizontally flipped to present the same leaf orientation as white light images.

The following figure supplements are available for figure 7:

**Figure supplement 1**. Western blots showing expression of AVR-Pik alleles, AVR-PikD mutants (**A**) and Pikp-1, Pikp-2 (**B**) in *N. benthamiana*.

**Figure supplement 2**. Box plots showing HR Index for repeats of the assay depicted in *Figure 7*.

amino acid polymorphism is sufficient to prevent AVR-Pik-dependent cell death in *N. benthamiana*, limit resistance in rice, and reduce the affinity of interaction with Pikp-HMA in vitro by an order of magnitude. This suggests a binding threshold is required to elicit a response in plants and the observed 10-fold reduced affinity of AVR-PikE for Pikp-HMA in vitro is sufficient for this. While this may be enough to explain why the conservative His46Asn mutation is the only one found in nature, it remains possible that other mutations would not be tolerated due to a trade-off with the effector's virulence activity.

## Mutation in other AVR-PikD interfacing residues also prevents Pikp-HMA binding

Of the other amino acid mutations explicitly designed to disrupt the Pikp-HMA/AVR-PikD interaction, AVR-PikD$^{Arg64Ala}$ and AVR-PikD$^{Asp66Arg}$ prevented interaction in SPR and yeast, and responses *in planta*. These results, in addition to AVR-PikD$^{His46Glu}$, provide convincing evidence that the crystal structure of Pikp-HMA/AVR-PikD is consistent with the complex formed in plant cells. Interestingly, AVR-PikD$^{Ile49Glu}$ retained interaction in SPR (threefold reduction in affinity compared to AVR-PikD) and yeast and elicited a response by Pikp *in planta*. Close inspection of the Pikp-HMA/AVR-PikD interface reveals how a Glu could be accommodated at this position. Of note is that AVR-PikD$^{Ile49}$ is

close to the N-terminus of the Pikp-HMA domain and a repositioning of Pikp-HMA$^{Gly186}$, for example in the context of the full-length protein, could create space for a Glu residue.

## Additional naturally occurring AVR-Pik polymorphisms also perturb binding to Pikp-HMA

AVR-PikA and AVR-PikC combine the AVR-PikD$^{His46Asn}$ polymorphism with a Pro47Ala/Gly48Asp substitution and an Ala67Asp substitution respectively (*Figure 1A*). AVR-PikC did not show measureable binding to Pikp-HMA in vitro or in yeast, or elicit responses *in planta*. In the Pikp-HMA/AVR-PikD structure, the side chain of AVR-PikD$^{Ala67}$ lies adjacent to the Pikp-HMA$^{Asp224}$/AVR-PikD$^{Arg64}$ salt-bridge. An Asp at position 67 would likely perturb the Pikp-HMA/AVR-PikD interaction through disruption of this salt-bridge. While reducing the affinity to a level where a $K_d$ cannot be determined in our SPR assay, and not eliciting a response in *N. benthamiana*, AVR-PikD$^{Ala67Asp}$ still interacted with Pikp-HMA in yeast and was recognised in rice when the protein was delivered by *M. oryzae*. It is possible that the weaker AVR-PikD$^{Ala67Asp}$/Pikp-HMA interaction may be sufficient for generating a response in rice containing Pikp, but not sufficient in the *N. benthamiana* assay used here.

AVR-PikA shows 23-fold apparent weaker affinity to Pikp-HMA compared to AVR-PikD in vitro, no interaction with Pikp-HMA in yeast, and elicited no response *in planta*. Despite being adjacent to AVR-PikD$^{His46}$, in the Pikp-HMA/AVR-PikD complex the side chains of AVR-PikD$^{Pro47}$ and AVR-PikD$^{Gly48}$ do not contact the same Pikp-HMA monomer and Ala/Asp residues could be accommodated at these positions without disrupting the Pikp-HMA/AVR-Pik interface. Consistent with this, the AVR-PikD$^{Pro47Ala/Gly48Asp}$ mutant showed only slightly reduced affinity for Pikp-HMA in vitro (2.7-fold), interacted with Pikp-HMA in yeast, and elicited responses *in planta*. This data suggests that AVR-PikD$^{Pro47Ala/Gly48Asp}$ mutations may provide a minor additive benefit, but AVR-PikD$^{His46Asn}$ is the dominant mutation contributing to the evasion of Pikp recognition. It is also plausible that these mutations contribute to the virulence activity of the effector in the context of an Asn at position 46, rather than quantitatively contributing to the evasion of Pik recognition.

## Understanding Pik-HMA specificity for AVR-Pik

Almost all amino acid variation between Pik-1 alleles lies within the HMA domain (*Figure 3—figure supplement 4B*). Mapping the variation in Pik-HMAs at the Pikp-HMA/AVR-PikD interface suggests a 'hot-spot' centred on recognition of AVR-Pik$^{46}$. Pikp-1$^{Glu230}$ (Pikp numbering), that directly co-ordinates AVR-PikD$^{His46}$, is a Val in both Pikm-1 and Pik*-1. Residue Pikp-1$^{Val222}$, whose side-chain extends towards Pikp-1$^{Glu230}$, is an Ala in Pikm-1 and Pik*-1. In the absence of specific binding data, none of these mutations in their own right, or when combined, would be predicted to preclude interaction with AVR-PikD (as has been observed previously [*Kanzaki et al., 2012*]), but would also not be predicted to explicitly enhance the binding of an Asn at position 46 (as found in AVR-PikE, AVR-PikA and AVR-PikC, which are all recognised by Pikm).

Three other Pik variable residues map at or close to the Pikp-HMA/AVR-PikD interface. Pikp-1$^{Asp217}$ is adjacent to the invariant Ser218 (which with Pikp-1$^{Glu230}$ co-ordinates AVR-PikD$^{His46}$) and extends towards residue 48 of AVR-PikD. It is conceivable that a His residue at position 217, as found in Pikm-1 and Pik*-1, may interact with residues in this region to extend recognition of AVR-Pik alleles, in particular Asp48 as found in AVR-PikA. Pikp-1$^{Lys228}$, a Glu in Pik*-1 and a Gln in Pikm-1, and Pikp-1$^{Glu253}$, a Met in Pikm-1, form hydrogen bonds with AVR-PikD$^{Asp66}$ and AVR-PikD$^{Lys79}$ respectively. Interestingly, position 228 is one of two residues that have been suggested as diagnostic markers for Pik breeding in rice (*Costanzo and Jia, 2010*). Despite the variation in these Pik residues, as Asp66 and Lys79 are invariant in AVR-Pik alleles, it is difficult to appreciate how they contribute to recognition specificity.

Teasing apart these seemingly fine-tuned recognition specificities between Pik-HMA and AVR-Pik alleles, and how these are balanced against the virulence-associated activity of the effectors, awaits further study.

## A model for activation of rice NLRs containing HMA domains

Recognition of AVR-Pik by Pik, and AVR-Pia and AVR1-CO39 by RGA5/RGA4, is by direct binding to the HMA domains of Pik-1 and RGA5. The position of the HMA domain, between the CC and NB-ARC region of Pik-1 and after the LRR in RGA5, is one of the most striking differences between these functionally-related proteins. Interestingly, the integrated domain in the Arabidopsis NLR RRS1,

a domain with sequence similarity to WRKY transcription factors, is also positioned after the LRR. Cognate effectors AvrRps4 and PopP2 directly interact with this WRKY domain (*Cesari et al., 2014*; *Le Roux et al., 2015*; *Sarris et al., 2015*). Together, these proteins reveal that functional integrated domains can occupy different positions in NLRs.

How does the binding of effectors to HMA domains trigger immunity-related signalling? The absence of obvious enzymatic activity in the effectors, and a lack of large conformation changes in Pikp-HMA in the AVR-PikD-bound and unbound states, supports the hypothesis that effector binding promotes domain re-arrangements in NLR complexes (*Cesari et al., 2014*). This could be by (1) direct competition for a shared binding surface on the HMA between the effectors and either an intra- or inter-molecular contact with another NLR domain, (2) the effectors may 'bridge' contacts between HMAs and other NLR domains to stabilise interactions, (3) effector binding disrupts or promotes interaction of NLRs with other, as yet unknown, molecules, (4) subtle changes within the HMAs, in the context of the full length proteins, promotes NLR domain rearrangements. Any of these scenarios could break existing and/or promote new interactions within NLR complexes. They could also promote presentation of new molecular surfaces that could interact with downstream components to initiate immunity-related signalling.

Due to the different positions of the Pik-1 and RGA5 HMA domains, the conformational changes underlying transduction of direct effector binding to immunity-related signalling are likely to be different, but the intra- and/or inter-molecular complexes mediating output maybe conserved. In the future, transferring unconventional integrated domains to the different positions within and between NLRs will determine the importance of domain location, and whether these positions can accommodate novel integrated domains with the potential to deliver new-to-nature resistance capabilities.

## Materials and methods

### Gene cloning: heterologous protein production, Y2H, fungal transformation and *in planta* expression

#### For protein production in insect cells and *E. coli*

DNA encoding the Pikp-HMA domain (residues Gly186 to Ser258, codon optimized for expression in *E. coli*) was synthesized and supplied in the pOPINF vector (*Berrow et al., 2007*) by Genscript (Pistcataway, NJ, United States). For sub-cloning into pOPINS3C (*Bird, 2011*), the Pikp-HMA sequence was amplified from the pOPINF vector above using primers shown in *Supplementary file 1*, followed by In-Fusion cloning (Clontech, Mountain View, CA, United States) with Kpn1/HindIII cut pOPINS3C. The resulting construct supports expression of a 6xHis+SUMO tagged Pikp-HMA domain linked by a 3C protease cleavage site.

DNA encoding AVR-Pik alleles AVR-PikD, E, A and C (residues Glu22 to Phe113, lacking the signal peptide), codon optimized for expression in *E. coli*, were synthesized and supplied in the pOPINF vector by Genscript. Each allele was sub-cloned into pOPINS3C using primers shown in *Supplementary file 1*, and In-Fusion cloning as described for Pikp-HMA above. Each of the four alleles were also cloned into pOPINE (*Berrow et al., 2007*) to facilitate expression of protein with a non-cleavable 6xHis tag on the C-terminus. To promote soluble protein expression, DNA encoding SUMO+AVR-Pik was amplified from pOPINS3C:AVR-Pik, using primers shown in *Supplementary file 1*, prior to insertion into pOPINE using In-Fusion cloning. A construct of AVR-PikD was also generated in pOPINA, using primers shown in *Supplementary file 1*, to enable production of protein without a tag and in a vector backbone compatible with co-transformation with other pOPIN vectors.

DNA encoding mutants of AVR-PikD (His46Glu, Ile49Glu, Arg64Ala, Asp66Arg, Ala67Asp, Pro47Ala/Gly48Asp), codon optimized for expression in *E. coli*, were synthesized and supplied in pDONR221 vector by Genscript (USA). All the mutants were sub-cloned into pOPINS3C (to deliver protein with a cleavable N-terminal 6xHis+SUMO tag) and pOPINE (to deliver protein with a cleavable N-terminal SUMO tag and non-cleavable C-terminal 6xHis tag) as described above.

#### For Y2H

DNA encoding AVR-PikD, E, A and C (residues Glu22 to Phe113, lacking the signal peptide and optimised for *E. coli* expression), and the Pikp-HMA domain (residues Gly-186 to Ser-258), were supplied by Genscript in Gateway entry vector pDONR221. AVR-Pik alleles, and the Pikp-HMA domain, were sub-cloned into destination vectors pDEST32 and pDEST22 respectively using LR

clonase (Life Technologies, United Kingdom). The AVR-PikD mutants were prepared by Genscript and supplied in the pDONR221 vector and transferred into pDEST32 as above.

### For *M. oryzae* transformation

All AVR-PikD mutants, with XbaI and EcoRI recognition sites added at the 5′ and 3′ ends, were produced and supplied by Genscript in the pUC57 vector. To generate each pCB1531:AVR-Pik (promoter)-AVR-PikD mutant (H46E, I49E, R64A, D66R, A67D, P47A/G48D), all pUC57:AVR-PikD mutants were digested with XbaI and EcoRI, and inserts were exchanged to the *mCherry* gene at the same sites of pCB:AVR-Pik(promoter)-AVR-Pik-mCherry (*Sharma et al., 2013*).

### For agroinfiltration assays in *N. benthamiana*

An AscI Phosphorylated Linker (NEB, Ipswich, MA, United States) was ligated to the SmaI site of pCambia1300 (Marker Gene Technologies, Inc, Eugene, OR, United States), generating pCambia1300:AscI. Three exon fragments of *Pikp-1* were PCR amplified from rice cv. K60 genomic DNA with primers given in *Supplementary file 1*. PCR products of the exons 1 and 2 were mixed, and further PCR amplified with the primers IFPik1U2 and IFPikp1L0.6 using the mixed PCR products as template. The resulting PCR fragment and the *Pikp-1* exon 3 fragment were mixed, and cloned into the AscI-cut pCambia1300:AscI by In-Fusion multiple fragment cloning (Clontech) to generate pCambia-Pikp-1. To assemble pCambia-C-3xFLAG, a SpeI recognition site plus 3xFLAG sequence with a stop codon (TGA) was introduced after the PstI recognition site of pCambia1300 by In-Fusion cloning. The *Pikp-1* coding sequence (CDS) was amplified with primers IFagctPikp1U2 and IFSpeIPikp1L0 (*Supplementary file 1*), using pCambia-Pikp-1 as a template, and cloned into SacI/SpeI cut pCambia-C-3xFLAG to generate pCambia-Pikp1-3xFLAG by In-Fusion cloning.

A pCambia-C-3xMyc vector was generated using essentially the same method as described above for pCambia-C-3xFLAG. The Pikp-2 *CDS* was amplified from cDNA (derived from RNA of a rice cv. K60 leaf, single-stranded cDNA was synthesized using oligo(dT) primer and ReverTra Ace [Toyobo, Japan]) with the primers SacIPikp2U2 and SpeIPikp2L0 (*Supplementary file 1*). The PCR product was digested with SacI and SpeI, and introduced into pCambia-C-3xMyc, generating pCambia-Pikp2-3xMyc.

A construct encoding AVR-PikDns-HS (AVR-PikD without the signal sequence and with a C-terminal HA-StrepII epitope tag) was generated by a three-step PCR approach using primers BP31nsU2 and P31YPYDVL2, BP31nsU2 and AHAL2 then BP31nsU2 and BSAPDYAL1 (*Supplementary file 1*) with pCB1004-pex31-D as the original template (*Yoshida et al., 2009*). The final PCR product was digested with BamHI and cloned into pCambia1300 in sense direction following the CaMV35S promoter. Each AVR-Pik allele was cloned in a similar fashion (to generate AVR-PikEns-HS, -Ans-HS or–Cns-HS) but with the original template being pCB1531:*AVR-Pik*(promoter):*AVR-Pik-E, -A* or–*C* (*Yoshida et al., 2009*), final PCR primers BP31nsU2 and XSAPDYAL1 (*Supplementary file 1*) and final cloning via BamH1/Xba1. This same PCR approach was used to generate CaMV35S promoter-driven tagged constructs for each of the AVR-PikD mutants in pCambia1300, but with the appropriate Genscript-supplied pUC57 vector used as the template.

All final constructs used in this study were verified by DNA sequencing.

## Y2H analyses

The Proquest two-hybrid system (Life Technologies, United Kingdom) was used to detect protein–protein interactions, according to manufacturer instructions, with only minor modifications. Briefly, DNA encoding Pikp-HMA in pDEST22 was co-transformed with either the individual AVR-Pik alleles or the AVR-PikD mutants in pDEST32, into chemically competent *Saccharomyces cerevisiae* MaV203 cells. Single colonies grown on selection plates were resuspended in 100 μl H$_2$O and 2 μl were spotted on SC-Leu-Trp (as growth control) and SC-Leu-Trp-His+10 mM 3AT (His auxotrophy assay). Photographs of colonies on SC-Leu-Trp-His+10 mM 3AT plates were taken after incubation for 24 hr at 28°C and 16 hr at room temperature.

For the X-gal assay, 2 μl of resuspended cells were spotted on a Hybond N membrane (GE Healthcare) on a YAPD plate. After 24 hr the membrane was removed and place on top of 2 layers of 3 MM paper (GE Healthcare) soaked with 10 ml buffer Z supplemented with 15 mg X-gal in 100 μl dimethylformamide and 60 μl 2-mercaptoethanol. After incubation for 24 hr at 37°C the membrane was air dried and a picture taken. All pictures are representative of at least three experimental repeats, with consistent results.

To confirm protein expression in yeast, total protein extracts from transformed colonies were produced according to the urea/SDS method as described in the Clontech Yeast Protocols Handbook. Aliquots of 4 µl were separated by SDS-PAGE and transferred to PVDF membrane. Due to the high level of expression of the GAL-4-DB domain from the empty pDEST32, a 10 µl of a dilution 1:100 of the original extract was loaded on SDS-PAGE gel for this sample. Membranes were probed with anti-GAL4-DBD HRP-conjugated antibody (Santa Cruz Biotechnology, Dallas, TX, United States) and developed with a mix of 500 µl of SuperSignal West Pico Chemiluminescent Substrate and 800 µl of SuperSignal West Femto Maximum Sensitivity Substrate (Life Technologies) following standard procedures.

## Heterologous protein expression and purification, intact mass spectrometry

### Pikp-HMA

For crystallisation, pOPINS3C:Pik-HMA was expressed in 1 l cultures of sf9 cells, infected with 15 ml baculovirus. The cells were incubated at 26°C, with continuous shaking at 250 rpm, for 48 hr then harvested by centrifugation. Cells were resuspended in 50 mM Tris HCl pH 7.5, 500 mM NaCl, 30 mM imidazole and 0.2% Tween 20 (buffer A) supplemented with EDTA free protease inhibitor tablets and DNAse 1. The cells were lysed by cell disruptor at 30 kpsi and cell debris was removed by centrifugation. The clarified lysate was applied to a $Ni^{2+}$-NTA column connected to an AKTA Xpress purification system. 6xHis+SUMO-Pikp-HMA was step-eluted with elution buffer (buffer A containing 500 mM imidazole) and directly injected onto a Superdex 75 16/60 gel filtration column pre-equilibrated in 20 mM Tris pH 7.5, 200 mM NaCl and 1 mM TCEP (buffer B). Fractions containing 6xHis+SUMO-Pikp-HMA (as assesses by SDS-PAGE) were pooled and concentrated to 2–3 mg/ml. The 6xHis-SUMO tag was cleaved by addition of 3C protease (10 µg/mg fusion protein) with overnight incubation at 4°C. Cleaved Pikp-HMA was purified from the digest using a $Ni^{2+}$-NTA column, collecting the eluate, followed by dialysis in buffer B and was then concentrated as appropriate.

For in vitro binding studies, pOPINS3C:Pikp-HMA was produced in *E. coli* SHuffle cells (*Lobstein et al., 2012*). Cell culture was grown in auto induction media (*Studier, 2005*) at 30°C for 24 hr and cells were harvested by centrifugation. Pelleted cells were resuspended in 50 mM Tris HCl pH 8, 500 mM NaCl, 50 mM Glycine, 5% (vol/vol) glycerol and 20 mM imidazole (buffer C) supplemented with EDTA free protease inhibitor tablets and lysed by sonication. The clarified cell lysate was applied to a $Ni^{2+}$-NTA column connected to an AKTA Xpress system. 6xHis+SUMO-Pikp-HMA was step-eluted with elution buffer (buffer C containing 500 mM imidazole) and directly injected onto a Superdex 75 26/600 gel filtration column pre-equilibrated in buffer D (20 mM HEPES pH 7.5 and 150 mM NaCl). The fractions containing 6xHis+SUMO-Pikp-HMA were pooled and concentrated to 2–3 mg/ml. The 6xHis+SUMO tag was cleaved by addition of 3C protease (10 µg/mg fusion protein) and incubation overnight at 4°C. Cleaved Pikp-HMA was further purified using a $Ni^{2+}$-NTA column (collecting the eluate) followed by gel filtration as above. The concentration of protein was judged by absorbance at 280 nm (using a calculated molar extinction coefficient of Pikp-HMA, 1400 $M^{-1}cm^{-1}$).

### AVR-Pik

For production of AVR-Pik alleles and mutants, relevant pOPIN constructs were introduced into SHuffle cells. Transformed SHuffle cells were grown in auto induction media, processed, and the proteins purified to homogeneity as described for Pikp-HMA above. The concentration of protein was judged by absorbance at 280 nm (using a calculated molar extinction coefficient for the relevant construct).

### Pikp-HMA/AVR-Pik complex

For crystallisation, Pikp-HMA and AVR-PikD were co-expressed in SHuffle cells following co-transformation with pOPINS3C:Pikp-HMA and pOPINA:AVR-PikD. Growth in the presence of carbenicillin and kanamycin maintained selection for both plasmids. Cells were grown in auto induction media, harvested and the protein sample was purified as described for Pikp-HMA. The concentration of protein was judged by absorbance at 280 nm (using a calculated molar extinction coefficient assuming a 2:1 complex of Pikp-HMA/AVR-PikD, 26,286 $M^{-1}cm^{-1}$).

### Intact mass spectrometry analyses

Protein intact masses were determined by LC-MS on a Synapt G2 mass spectrometer coupled to an Acquity UPLC system (Waters, United Kingdom). 50–100 pmol of protein were injected onto an Aeris

WIDEPORE 3.6 µ C4 column (Phenomenex, United Kingdom) and eluted with a 10–90% acetonitrile gradient over 13 min (0.4 ml/min). The spectrometer was controlled by the Masslynx 4.1 software (Waters) and operated in positive MS-TOF and resolution mode with capillary voltage of 2 kV, cone voltage, 40 V. Leu-enkephalin peptide (2 ng/ml, Waters) was infused at 10 µl/min as a lock mass and measured every 30 s. Spectra were generated in Masslynx 4.1 by combining scans and deconvoluted using the MaxEnt1 tool (Waters).

## Protein:protein interaction studies in solution

### Analytical gel filtration

Analytical size exclusion chromatography was performed at 4°C using a Superdex 75 10/300 gel filtration column (GE Healthcare) pre-equilibrated in 50 mM HEPES pH 7.5 and 150 mM NaCl. Samples were centrifuged prior to loading. A 100 µl of the sample was injected at a flow rate of 0.8 ml/min and 0.5 ml fractions were collected for analysis by SDS-PAGE gels. To study complex formation, proteins were mixed and incubated on ice for 60 min prior to loading.

### Surface plasmon resonance

SPR experiments were performed at 25°C using a Biacore T200 system (GE Healthcare). For interaction studies proteins were prepared in buffer E (50 mM HEPES pH 7.5, 150 mM NaCl and 0.1% Tween 20) and all the measurements were recorded using buffer E at a flow rate of 30 µl/min. All experiments were performed using an NTA sensor chip (GE Healthcare). A multi-cycle kinetics approach was used to study interaction between Pikp-HMA and four alleles of effector protein. For each cycle the chip was activated by injecting 30 µl of 0.5 mM $NiCl_2$ over flow cell 2 and His-tagged protein (AVR-Pik) was immobilised on flow cell 2 until a response level of 250 ± 10 was acheived. Different concentrations of Pikp-HMA (ranging from 1 to 1200 nM) and buffer only controls were injected over flow cells 1 and 2 (flow cell 1 was used as reference) for 120 s and dissociation was recorded for another 300 s. Binding responses were recorded at each concentration of Pikp-HMA just before the end of injection and these were then fitted to a steady state affinity model assuming 1:1 binding. The inclusion of buffer-only controls enabled the use of double referencing whereby for each analyte measurement, in addition to subtracting the response in FC 1 from the response in FC 2, a further buffer-only subtraction was made to correct for bulk refractive index changes or machine effects (*Myszka, 1999*). Interaction studies of Pikp-HMA and AVR-PikD mutants were performed using a single cycle kinetics method. The chip was activated by injecting 30 µl of 0.5 mM $NiCl_2$ over FC 2 and was used to immobilize His-tagged protein (AVR-PikD mutants) on flow cell 2 to a response level of 250 ± 10. Increasing concentrations of Pikp-HMA (1, 10, 20, 60 and 120 nM) were injected over flow cell 1 and 2 for 120 s. After the final injection the dissociation was recorded for 300 s. Two startup cycles were run where the chip was activated and effector proteins immobilised in the same manner, but buffer only was injected. This was subtracted to account for any dissociation of AVR-PikD mutants from the sensor chip. For both types of kinetic experiments the sensor chip was regenerated by injecting 30 µl of 0.35M EDTA. All the data were analyzed using Biacore T200 BiaEvaluation software (GE Healthcare). The raw data was exported and plotted using Microsoft Excel. Each experiment was done in duplicate, with similar results.

## Crystallisation, data collection, structure determination and refinement

### Pikp-HMA

For crystallization, Pikp-HMA was concentrated to 6 mg/ml in 20 mM Tris, pH 7.5, 200 mM NaCl and 1 mM TCEP. Crystallization experiments were performed using an Oryx nano robot (Douglas Instruments, United Kingdom) and sitting drop vapor diffusion in 96 well plates. Pikp-HMA produced crystals after 24–36 hr in 0.1 M MIB buffer, pH 5.0 and 25% PEG 1500 (PACT screen, Molecular Dimensions, United Kingdom). For X-ray data collection, crystals were transferred to the precipitant solution with the addition of 20% ethylene glycol (as a cryoprotectant), mounted in a litho loop and flash cooled in liquid nitrogen. Pikp1-HMA also produced crystals in another condition of the PACT (0.2 M potassium thiocyanate and 20% PEG 3350). These crystals were soaked for 45 s in well solution supplemented with 300 mM potassium iodide and were cryoprotected as above prior to freezing in liquid nitrogen.

Native and SAD (single wavelength anaomalous diffraction) X-ray data sets were collected from Pikp-HMA crystals at the Diamond Light Source, United Kingdom beamline I04. The data were

processed using the Xia2 pipeline (*Winter, 2010*). The structure was solved using the SAD approach with the data collected from the crystal soaked in iodide, and the AutoSol wizard as implemented in PHENIX (*Adams et al., 2010*), which also built an initial model. The final structure was obtained through iterative cycles of manual rebuilding and refinement using COOT (*Emsley et al., 2010*) and REFMAC5, as implemented in CCP4 (*Winn et al., 2011*), using the Native data. Structure validation used the tools provided in COOT and MOLPROBITY (*Chen et al., 2010*).

### Pikp-HMA/AVR-PikD complex

Crystals of Pikp-HMA/AVR-PikD were grown using purified complex concentrated to 10 mg/ml and 0.2 M ammonium sulphate, 0.1 M CHES (N-Cyclohexyl-2-aminoethanesulfonic acid) pH 9 and 20% PEG3350. Crystals were cryoprotected with mother liquor containing 20% ethylene glycol, then mounted in a Litho loop and flash cooled in liquid nitrogen. X-ray diffraction data were collected at the Diamond Light Source, beamline I04.

For structure solution, the model of Pikp-HMA was positioned in the asymmetric unit of the Pikp-HMA/AVR-PikD crystal by molecular replacement using Phaser (*McCoy et al., 2007*). The phased data was density modified using PARROT (*Cowtan, 2010*). The resulting data were used in BUCCANEER (*Cowtan, 2006*), which was able to build approximately half of the AVR-PikD structure, in addition to most of the Pikp-HMA dimer, with just the protein sequences supplied. From this point the final structure was completed through iterative rounds of manual rebuilding, refinement and validation as described previously.

## Pathogenicity assays

*M. oryzae* strains Sasa2, Sasa2 with pex31-D fragment (AVR-PikD) or with AVR-PikE used in this study are stored at the Iwate Biotechnology Research Center (*Yoshida et al., 2009*; *Kanzaki et al., 2012*). To obtain protoplasts, hyphae of each Sasa2 strain were incubated for 3 days in 200 ml of YG medium (0.5% yeast extract and 2% glucose, wt/vol). Protoplast preparation and transformation were performed as described previously (*Takano et al., 2001*). Bialaphos-resistant transformants were selected on plates with 250 µg/ml of bialaphos (Wako Pure Chemicals).

Rice leaf blade spot inoculations were performed with *M. oryzae* strains (*Kanzaki et al., 2002*). Disease lesions were photographed 14 days post inoculation. Rice seedlings (cvs. Nipponbare and K60) at the fourth leaf stage were used for inoculation. The assays were repeated at least 3 times with similar results.

For RT-PCR, total RNA was extracted from disease lesions of rice cv. Nipponbare leaves using Purelink Plant RNA Reagent, which was subsequently treated with TURBO DNase (Life Technologies). From 2 µg of the DNase-treated RNA of each sample, single-strand cDNA was synthesized using oligo (dT) primer and ReverTra Ace (Toyobo). To confirm the gene expression of *AVR-Pik* and the *M. oryzae* actin gene (*Mo-Actin*), these genes were amplified by PCR with primers given in (*Supplementary file 1*).

## *In planta* expression to monitor cell death

For agroinfiltration in *N. benthamiana*, *A. tumefaciens* strain GV3101 was transformed with the relevant binary constructs. Leaves of 4 weeks old *N. benthamiana* plants were agroinfiltrated using a needleless syringe. The total $OD_{600}$ of infiltrated cultures was 1.0 with ratios used 1.5:1.5:6:1 for Pikp-1:Pikp-2:effector:P19. When one or more constructs were not present, total $OD_{600}$ was maintained with appropriate amount of empty vector. Photos showing cell death were taken 4 dpi from the adaxial side of the leaves for white light images and abaxial side of the leaves for UV images. Pictures are representative of four independent experiments, with internal repeats. Data for the box plots presented in *Figure 6—figure supplement 4* and *Figure 7—figure supplement 2* are from three independent experiments with internal repeats. The HR index was scored according to the scale presented in *Figure 6—figure supplement 3*.

For extraction of total protein from samples, leaf disks were taken at 2 dpi and homogenised in extraction buffer (250 mM Tris–HCl, pH 7.5, 2.5 mM EDTA, 0.1% ascorbic acid, 1 mM PMSF, 0.1% [vol/vol] Protease Inhibitor Cocktail for plant cell and tissue extracts [SIGMA, St. Louis, MO, United States]). Supernatants were centrifuged and separated on 10–20% precast e-PAGEL gels prior to transfer onto Immobilon Transfer Membranes (Millipore, Germany). The blots were blocked in 2% ECL Advance Blocking Agent (GE Healthcare) in TTBS (10 mM Tris–HCl, pH 7.5, 100 mM NaCl, 0.1% Tween 20 [vol/

vol]) for 1 hr at room temperature with gentle agitation. For immunodetection, blots were probed with anti-HA (3F10)-HRP (Roche, Switzerland), anti-FLAG M2-HRP (SIGMA) or anti-Myc-tag (HRP-DirecT) (MBL, Woburn, MA, United States) in a 1:10,000 dilution in TTBS for 2 hr. After washing the membrane for 3 × 10 min, the reactions were detected using ChemiLumi One Super or Ultra (Nacalai Tesque, Japan) and a Luminescent Image Analyzer LAS-4000 (Fujifilm, Japan).

## Accession codes

Protein structures, and the data used to derive these, have been deposited at the PDB with accession numbers 5a6p (Pikp-HMA), 5a6w (Pikp-HMA/AVR-PikD complex).

## Acknowledgements

For the UK, this work was supported by the BBSRC (grant BB/J00453); the ERC proposal 'NGRB'; the John Innes Foundation and the Gatsby Charitable Foundation. For Japan, this work was supported by the Programme for Promotion of Basic and Applied Researches for Innovations in Bio-oriented Industry, Grant-in-aid for MEXT (Scientific Research on Innovative Areas 23113009) and JSPS KAKENHI (Grant Nos. 24248004, 26292027, 15H05779). We gratefully acknowledge the Diamond Light Source (UK) for access to X-ray data collection facilities and expert support and the OPPF (UK) for help with protein production (in particular, Pikp-HMA expressed in insect cells). We also thank Gerhard Saalbach for mass spectrometry and all members of the Banfield Laboratory for discussions.

## Additional information

### Funding

| Funder | Grant reference | Author |
| --- | --- | --- |
| Biotechnology and Biological Sciences Research Council (BBSRC) | BB/J00453 | A Maqbool, M Franceschetti, CEM Stevenson, S Kamoun, MJ Banfield |
| Biotechnology and Biological Sciences Research Council (BBSRC) | BB/M02198X | M Franceschetti, MJ Banfield |
| John Innes Foundation (JIF) | | A Maqbool, M Franceschetti, CEM Stevenson, MJ Banfield |
| Gatsby Charitable Foundation | | S Kamoun |
| Japan Society for the Promotion of Science (JSPS) | 24248004, 26292027, 15H05779 | H Saitoh, A Uemura, H Kanzaki, R Terauchi |
| European Research Council (ERC) | Proposal NGRB | A Maqbool, S Kamoun, MJ Banfield |

The funders had no role in study design, data collection and interpretation, or the decision to submit the work for publication.

### Author contributions

AM, HS, Conception and design, Acquisition of data, Analysis and interpretation of data, Drafting or revising the article; MF, Acquisition of data, Analysis and interpretation of data, Drafting or revising the article; CEMS, AU, HK, Acquisition of data, Analysis and interpretation of data; SK, Conception and design, Drafting or revising the article; RT, MJB, Conception and design, Analysis and interpretation of data, Drafting or revising the article

### Author ORCIDs

S Kamoun, http://orcid.org/0000-0002-0290-0315
MJ Banfield, http://orcid.org/0000-0001-8921-3835

## Additional files

### Supplementary file

• Supplementary file 1. Excel spreadsheet. Primers used in the construction of expression vectors.

### Major datasets

The following datasets were generated:

| Author(s) | Year | Dataset title | Dataset ID and/or URL | Database, license, and accessibility information |
|-----------|------|---------------|----------------------|-------------------------------------------------|
| Maqbool A, Saitoh H, Franceschetti M, Stevenson CE, Uemura A, Kanzaki H, Kamoun S, Terauchi R, Banfield MJ | 2015 | Heavy metal associated domain of NLR-type immune receptor Pikp1 from rice (Oryza sativa) | http://www.rcsb.org/pdb/search/structidSearch.do?structureId=5A6P | Publicly available at the RCSB Protein Data Bank (accession no. 5A6P). |
| Maqbool A, Saitoh H, Franceschetti M, Stevenson CE, Uemura A, Kanzaki H, Kamoun S, Terauchi R, Banfield MJ | 2015 | Complex of rice blast (Magnaporthe oryzae) effector protein AVR-PikD with the HMA domain of Pikp1 from rice (Oryza sativa) | http://www.rcsb.org/pdb/search/structidSearch.do?structureId=5a6w | Publicly available at the RCSB Protein Data Bank (accession no. 5a6w). |

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
