## [Decision Letter]

Thank you for submitting your work entitled “Structural basis of pathogen recognition by a sensor domain in a plant NLR immune receptor” for peer review at *eLife*. Your submission has been favorably evaluated by Detlef Weigel (Senior Editor) and three reviewers, one of whom, Thorsten Nürnberger, is a member of our Board of Reviewing Editors. The reviewers have discussed their reviews with one another, and the Reviewing Editor has drafted this decision to help you prepare a revised submission.

All reviewers agree that you provide an excellent and very thorough paper that adds significantly to our understanding of how plant NLR receptors recognize pathogen effector proteins in the first step of their activation. NLR activation, still poorly understood at the mechanistic level, leads to successful plant immune response and thus halts pathogen growth. This topic is timely, important and therefore suitable for the *eLife* readership. In particular, the following findings make this work a valuable contribution to the field of plant immunobiology:

a) The demonstration of physical interaction of a fungal effector with an NLR receptor subunit.

b) The quantitative analysis of this interaction, which provides evidence that affinities of effectors to NLR immune receptors are in the same range as ligand affinities of AvrPto to Pto and microbial patterns to their cognate plant PRRs.

c) Comparative mutational analyses that allow to pinpoint high-affinity interactions to individual amino acid residues and thus to deduce a molecular mechanism for immune receptor activation.

There are, however, a number of concerns you might want to address in a revised submission:

1) One concern deals with the stoichiometry of the receptor–ligand complex that was determined as 2:1. You state however that it is mostly a 1:1 ratio between the NLR and the effector that is required for biological function. Is there any steric limitation that doesn't allow a second AVR-PikD to interact with the other HMA monomer thereby yielding a heterotetrameric α_2_ß_2_ complex?

2) Another concern is that the authors did not make sufficiently clear what the role of Pikp-2 in AvrPikD-mediated immune activation is and whether or not they could exclude heteromeric complex formation with Pikp-1 in an effector-mediated manner. That could be tested by using recombinant Pikp1 and Pikp2 in the presence and absence of effector in vitro.

3) As discussed in the manuscript, N46 in AVR-PikE is a polar residue and in principle can form hydrogen bonds with S218 and E230 of HMA. Please provide an explanation for why AVR-PikE has a much weaker affinity with HMA than AVR-PikD.

4) The 'integrated sensor' vs 'integrated decoy' domain nomenclature does no one outside the field any good in trying to understand how these systems works. To keep using two opposed terms, decoy and sensor, is problematic, especially since the NLR field already has 'sensor' and 'helper' NLRs and that this binary is specifically mapped onto NLR pairs, such as those described here. In this case, the NLR with the newly discovered 'integrated' domain is on the 'sensor NLR', which the paired NLR uses to activate disease resistance. It seems by the authors' terminology that the 'sensor NLR' is a 'sensor integrated domain-containing sensor NLR'. Here, clarification is needed.

While the arguments that attended the start of this discussion among the experts in this particular field are appreciated, it needs to be demonstrated that, in this particular case, the HMA fusion domain is not a decoy. In fact, as noted, the HMA fusion domain lacks metal binding capabilities required for intrinsic HMA function and is thus most likely a decoy domain.

---

## [Author Response]

*1) One concern deals with the stoichiometry of the receptor-ligand complex that was determined as 2:1. You state however that it is mostly a 1:1 ratio between the NLR and the effector that is required for biological function. Is there any steric limitation that doesn't allow a second AVR-PikD to interact with the other HMA monomer thereby yielding a heterotetrameric α*_*2*_*ß*_*2*_
*complex?*

The reviewers’ conclusion is correct. Inspection of the 2:1 complex structure determined here reveals a second AVR-Pik molecule cannot interact with the other HMA domain in the dimer (with the same interaction surface) due to a significant steric clash. This clash predominantly involves AVR-Pik residues 42–48, but also the C-terminus of the effector. It is not possible for a heterotetrameric α_2_ß_2_ complex to form.

We have added the following sentence to the manuscript to clarify this point: “Further, due to steric clash that would occur, it is not possible for an AVR-PikD/Pikp-HMA heterotetramer (2:2 complex) to assemble”.

*2) Another concern is that the authors did not make sufficiently clear what the role of Pikp-2 in AvrPikD-mediated immune activation is and whether or not they could exclude heteromeric complex formation with Pikp-1 in an effector-mediated manner. That could be tested by using recombinant Pikp1 and Pikp2 in the presence and absence of effector* in vitro*.*

In this manuscript we focus on dissecting the direct interaction event between effector and NLR in the AVR-Pik/Pik system (effector binding to the HMA domain of Pik-1). Our in planta experiments show that Pikp-1, Pikp-2 and AVR-PikD together are required for cell death. At this time we do not know the precise role of Pikp-2 in effector-mediated immune activation and can only speculate on this. As we state in the manuscript, we have yet to dissect the pre- and post-activation complexes in the AVR-Pik/Pik system and we cannot confirm or exclude heteromeric complex formation between Pikp-1 and Pikp-2 in the presence or absence of effectors. To further emphasise these points, we have added the following sentence: “At present it is unknown whether or not Pikp-2 forms a heteromeric complex with Pikp-1 in an effector-dependent manner.”

Producing recombinant full-length NLR proteins, derived from animals or plants, for use in in vitro experiments is very challenging and we have yet to attempt this with Pikp-1 or Pikp-2. We will work towards this in the future.

*3) As discussed in the manuscript, N46 in AVR-PikE is a polar residue and in principle* can *form hydrogen bonds with S218 and E230 of HMA. Please provide an explanation for why AVR-PikE has a much weaker affinity with HMA than AVR-PikD*.

In the original manuscript we stated: “Conceptually, this residue could be accommodated at the Pikp-HMA/AVR-PikD interface without disruption to the interaction. However, this single amino acid…”. With hindsight, this sentence does not fully convey our intent in describing the impact of this variant. In our revision, we now state: “Conceptually, this residue could be accommodated at the Pikp-HMA/AVR-PikD interface without generating significant steric clashes, but the interactions formed at this site (e.g. hydrogen bonding pattern) will be fundamentally different. This single amino acid…”

There are many subtleties at interfaces that will dictate overall binding affinities between proteins. As yet, there are not any reliable computational approaches to predict the precise effects of mutations on protein:protein interactions. Inspection of the Pikp-HMA/AVR-PikD complex, with the His-to-Asn mutated in silico, confirms a very different predicted hydrogen-bonding pattern between the proteins. Further, shape complementarity in the binding pocket appears less ideal with Asn. Therefore, in the absence of a Pikp-HMA/AVR-PikE structure, we cannot precisely correlate the experimentally determined reduction in binding affinity of AVR-PikE to Pikp-HMA, compared to AVR-PikD, with specific individual molecular interactions at the interface at this time.

*4) The 'integrated sensor' versus 'integrated decoy' domain nomenclature does no one outside the field any good in trying to understand how these systems works. To keep using two opposed terms, decoy and sensor, is problematic, especially since the NLR field already has 'sensor' and 'helper' NLRs and that this binary is specifically mapped onto NLR pairs, such as those described here. In this case, the NLR with the newly discovered 'integrated' domain is on the 'sensor NLR', which the paired NLR uses to activate disease resistance. It seems by the authors' terminology that the 'sensor NLR' is a 'sensor integrated domain-containing sensor NLR'. Here, clarification is needed*.

While the arguments that attended the start of this discussion among the experts in this particular field are appreciated, it needs to be demonstrated that, in this particular case, the HMA fusion domain is not a decoy. In fact, as noted, the HMA fusion domain lacks metal binding capabilities required for intrinsic HMA function and is thus most likely a decoy domain.

We appreciate the comment of the reviewers. It is true that the available evidence would suggest that the integrated HMA domain in Pikp-1 is a decoy. However, the experiment suggested by Wu et al*.* (Frontiers in Plant Science 6: 134, 2015), a genetic test to inform whether an integrated domain in an NLR has retained a biochemical activity independent of perception of an avirulence effector, has yet to be performed. Therefore, in this revised manuscript we have used the agnostic term ‘integrated domain’ throughout, except in one place where we state that “…integrated domains in the NLRs (known as decoy or sensor domains) that…”. We hope that this is a satisfactory use of terminology and reflects the views of all experts in the field.